# Research on the Design of Coal Mine Microseismic Monitoring Network Based on Improved Particle Swarm Optimization

**Kaikai Wang** [1,2], **Chun'an Tang** [1,2], **Ke Ma** [1,2,*] and **Tianhui Ma** [1,2,*]

1   State Key Laboratory of Coastal and Offshore Engineering, Dalian University of Technology, Dalian 116024, China
2   Institute of Rock Instability and Seismicity Research, Dalian University of Technology, Dalian 116024, China
*   Correspondence: mark1983@dlut.edu.cn (K.M.); tianhuima@dlut.edu.cn (T.M.)

**Abstract:** The quality of a mine's microseismic network layout directly affects the location accuracy of the microseismic network. Introducing the microseismic probability factor $F_e$, the microseismic importance factor $F_Q$, and the effective range factor $F_V$, an improved particle swarm algorithm with bacterial foraging algorithm is proposed to optimize the mine's microseismic network layout and evaluation system based on the $D$-value optimization design theory. Through numerical simulation experiments, it is found that the system has the advantages of fast optimization speed and good network layout effect. Combined with the system application at Xiashijie Coal Mine in Tongchuan City, Shaanxi Province, the method in this paper successfully optimizes the layout of the 20-channel network, ensuring that the positioning error of key monitoring areas is controlled within 20 m, and the minimum measurable magnitude can reach $-3.26$. Finally, it is verified by blasting tests that the maximum spatial positioning accuracy of the site is within 12.2 m, and the positioning capability of the site network is more accurately evaluated. The relevant research can provide a reference for the layout of the microseismic monitoring network for similar projects.

**Keywords:** microseismic network layout; the $D$-value optimization design theory; improved particle swarm algorithm (IPSO); numerical simulation experiment; blasting test

## 1. Introduction

With the increase in large-scale mining equipment and the improvement in mine production management levels, long-term mining leads to the depletion of shallow mineral resources and the continuous increase in mining depth [1,2]. Due to the existence of complex environments such as deep high stress, the incidence of high-energy rock bursts and mine earthquakes increases [3], and traditional monitoring methods are difficult to accurately analyze due to the main distribution of the surrounding rock surfaces [4]. Microseismic monitoring technology utilizes modern computing technology, communication technology, GPS timing, and precise positioning technology to determine the location and magnitude of microseismic events in rock mass in three-dimensional space. The technology catches microseismic events, which occur in the form of weak seismic waves in the process of fault failure, to perform a safety evaluation of the deformation activity and stability of the deep coal and rock mass [5].

The positioning accuracy of the source is an important indicator to measure the monitoring quality of the microseismic network, which determines the effect of the microseismic monitoring to a large extent [6]. The quality of the network layout directly affects the microseismic location accuracy [7,8]. Foreign scholars have conducted a lot of research in this area and achieved remarkable results. For example, Kijko [9,10] and Mendecki et al. [11] proposed a microseismic network evaluation method based on the optimal design theory of $D$ and $C$ values (determinant value of the source covariance matrix and the condition number of non-linear travel-time equations with respect to the known source parameter

vector). Although these two methods use numerical methods to evaluate the positioning capability of the network, they do not consider the key monitoring areas and actual engineering conditions. Andrzej [12] used the covariance matrix of the microseismic arrival equation to evaluate the positioning ability of the combined single-component sensor and three-component sensor arrangements, respectively. Zhang et al. [13] significantly improved the positioning error of the three-channel sensor arrangement by increasing the number of sensors reasonably. Tang et al. [14] used the $D$-value optimization method to study the optimization of the station network layout of the microseismic monitoring system, designed several spatial layout schemes for the sensor station network, and calculated the event source location error and system sensitivity for each scheme. Gong et al. [15,16] put forward the general principles for determining the risk monitoring area of rockburst and the layout requirements for station candidate points by using the comprehensive index method and then using the $D$-value optimization criterion to form the final scheme. Gao et al. [17] added relevant influencing factors to reconstruct the objective function of the network optimization based on the $D$-value theory, combined with the actual situation of mine engineering so that the microseismic monitoring system can meet the actual needs of the mine. Li et al. [18] used principal component analysis (PCA) to construct a comprehensive optimization analysis model for the microseismic monitoring network. However, these studies only solve the problems of monitoring range and monitoring accuracy calculation from the technical level. Due to the complexity of underground engineering, the arrangement of the microseismic monitoring network is greatly restricted, so it is often necessary to select multiple schemes while meeting the technical requirements. Moreover, the above methods give more certain evaluation indicators, and comprehensive network optimization determination and evaluation methods need to be developed and proposed.

In terms of scheme optimization, the traditional empirical analogy method is highly subjective, and it is difficult to achieve a quantitative judgment. For example, the exhaustive algorithm (EA) has a huge computational cost for problems with too many candidate points. With the development of optimization theory, many intelligent methods have been introduced into scheme optimization in recent years. Li et al. [19] proposed an improved multi-path immune particle swarm optimization based on the transport strategy (IPSMT) to optimize multi-path transmission routing in dynamic wireless sensor networks with movable nodes. Wang et al. [20] proposed an inertial weighted particle swarm optimization (NL-wPSO) algorithm based on nonlinear decreasing low-latency layout planning to optimize the low-latency planning problem of 5G networks. Su et al. [21] proposed the design optimization of the McPherson suspension system for minivans based on the weighted combination method and the neighborhood cultivation genetic algorithm. Liu et al. [22] established a systematic index evaluation model based on the fuzzy analytic hierarchy process model, considering the multi-indicator and multi-level structure of technology, science, and funding. Shen et al. [23] pointed out that the evaluation and optimization of mining schemes are greatly influenced by subjective factors, analyzed various factors required for the comprehensive index and multi-plan comparison and selection of uranium mining plans, and selected a scientific and reasonable evaluation. The weight of each index is optimized by introducing a genetic algorithm, and a mathematical model for the evaluation and optimization of the uranium mining scheme is constructed. However, traditional optimization algorithms easily fall into local extremums for solving discrete and complex problems. It is urgent to propose reasonable and improved intelligent optimization algorithms according to the complexity of actual problems.

Based on the above problems, the microseismic probability factor $F_e$, the microseismic importance factor $F_Q$, and the effective range factor $F_V$ are introduced according to the actual conditions of coal mines. The $D$-value optimization criterion optimized by the improved particle swarm algorithm (IPSO) forms the optimal plan for the microseismic network layout, and the evaluation indicators are proposed to characterize the sensitivity and positioning error and verify the superiority of the microseismic network layout through

simulation tests and field applications. Finally, a global optimal layout and evaluation system for a microseismic network layout is formed.

## 2. Optimization Theory of the Microseismic Network

The spatial array of sensors is one of the factors that affect the reliability of microseismic data. Considering the reasonable arrangement density, installation horizon, and other conditions, a small system positioning error can be guaranteed. Because of the above problems, the method of analyzing the positioning accuracy error is usually adopted to optimize the layout scheme of the combined system. The *D*-value theory holds that the size of the determinant of the covariance matrix of source parameters is proportional to the volume of the error ellipsoid. The method has been successfully applied to various fields [14,24].

The microseismic source is H $(t_0, x_0, y_0, z_0)$, and the $i^{th}$ station is S$_i$ $(t_i, x_i, y_i, z_i)$. For the uniform and isotropic velocity models, the shortest time $T_i$ from the source H to the $i^{th}$ station S$_i$ can be described by Equation (1):

$$T_i = \frac{\sqrt{(x_0 - x_i)^2 + (x_0 - x_i)^2 + (x_0 - x_i)^2}}{V_p} \tag{1}$$

In the equation, H $(t_0, x_0, y_0, z_0)$ are the time and three-dimensional coordinates of the microseismic source, respectively. S$_i$ $(t_i, x_i, y_i, z_i)$ are the time and three-dimensional coordinates of the $i^{th}$ sensor, respectively. $V_p$ is the uniform microseismic propagation velocity, $i = 1, 2, \ldots, n$, where $n$ is the number of stations installed in the mine. Kijko [9,10] considered that the optimization of the sensor station location depends on the covariance matrix $C_x$ of $x$, as shown in Equation (2):

$$C_x = k(A^T A)^{-1}$$
$$A = \begin{bmatrix} 1 & \frac{\partial T_1}{\partial x_0} & \frac{\partial T_1}{\partial y_0} & \frac{\partial T_1}{\partial z_0} \\ \vdots & \vdots & \vdots & \vdots \\ 1 & \frac{\partial T_n}{\partial x_0} & \frac{\partial T_n}{\partial y_0} & \frac{\partial T_n}{\partial z_0} \end{bmatrix} \tag{2}$$

In Equation (2), $A$ is the calculated partial differential matrix with the corresponding earthquake arrival time, and $k$ is a constant. This covariance can be graphically explained with the confidence ellipsoid, i.e., the eigenvalues of the covariance matrix constitute the length of the principal axis of the confidence ellipsoid. Finding the station arrangement with the smallest volume of the ellipsoid is called the optimal design of the $D$ value. The volume of the ellipsoid is proportional to the product of the covariance eigenvalues, that is, to the determinant of $C_x$. As shown in Equation (3), det $[C_x]$ is minimized to satisfy the $D$-value optimization criterion:

$$obj = \min \left( \sum_{i=1}^{ne} M_e(h_i) \lambda x_0(h_i) \lambda y_0(h_i) \lambda z_0(h_i) \lambda t_0(h_i) \right) \tag{3}$$

In the equation, *ne* is the number of hypocenter points calculated in the monitoring area. $M_e(h_i)$ is the microseismic event impact factor index. $\lambda_{x0}(h_i)$, $\lambda_{y0}(h_i)$, $\lambda_{z0}(h_i)$, and $\lambda_{t0}(h_i)$ are the eigenvalues of $C_x$.

## 3. Impact Factors Analysis

The microseismic network layout scheme is preferably a decision-making system project involving multiple factors, multiple indicators, and complex decision-making systems, and there are multiple correlations between the indicators and variables. When using these indicators for specific data analysis, the situation will be very complicated. The main factors involved the probability of microseismic events, positioning accuracy

requirements, effective monitoring range, monitoring area importance, construction condition requirements, economic factor indicators, and effective use time, etc. The goal of the *D*-value optimization criterion is to meet the positioning precision requirements. The construction conditions are mainly considered from the perspective of traffic and economy. It is necessary to investigate whether the site has the feasibility of installing geophones in boreholes. The installation location requires small interference signals, stable lithology, and no broken zone. Pre-selected feasible areas are made according to the site inspection. The economic factor indicators and the effective usage time should be evaluated and completed before equipment installation. The actual influencing factors that we need to care about include the probability of microseismic events, the effective monitoring range, and the importance of monitoring areas.

### 3.1. The Microseismic Probability Factor

There are many micro-fracture events in rock mass under mining disturbance, and the distribution law is complex. When microseismic monitoring is not carried out in mines, it is often assumed that the probability of microseismic occurrence in each area is the same to simplify the calculation. With the advancement of the mining face, when the geophone needs to be moved, the microseismic activity law during this period is used as a reference for the probability of microseismic events to optimize the next monitoring network layout scheme in the mining area. The microseismic probability factor in the whole mining area always satisfies Equation (4):

$$\int_{\Omega} F_e(H_i) dH_i \equiv 1 \tag{4}$$

This equation is based on analyzing the rockburst disaster in each region taking into account the geological and mining factors in the region. $F_e(H_i)$ is determined by numerical simulation and field measurement.

### 3.2. The Microseismic Importance Factor

In the process of underground mining, due to the complex and changeable ore body shape and mining conditions, the microseismic monitoring range is often of irregular geometry. The macroscopic requirement of network optimization is that the spatial geometry formed by the stations has good properties, that is, the effective monitoring range and the designed monitoring range are highly consistent [25]. To construct the monitoring range index of the station network from a quantitative point of view, it is necessary to define the effective monitoring range. The *D*-value method is used to calculate the theoretical positioning error under the network, and it is considered that the three-dimensional space with an error less than e (the value of e is set by the mine according to the needs of safety production, such as 50 m) is an effective monitoring area [17]. The design monitoring range is recorded as $V_0$, the effective monitoring range of the station network is $V_1$, and the overlapping area of the design monitoring range and the effective monitoring range is $V'$. The microseismic importance factor $F_V$ is introduced, and its mathematical definition is shown in Equation (5):

$$F_V = \frac{V'}{V_0} \tag{5}$$

Ideally, the effective monitoring range of the station network completely coincides with the designed monitoring range, that is, $F_V = 1$, and it is stipulated that $F_V \leq 1$.

### 3.3. The Effective Range Factor

The deployment of the microseismic network in the mining area is mainly to monitor the stability of the stope, control the large-scale subsidence and subsidence of the stope, protect the mineral resources and ecological environment, and prevent the occurrence of coal or rock dynamic disasters. Therefore, the dangerous area or special location of the stope should be designated as the key monitoring area. At the same time, it is necessary to monitor the disturbance impact of the current mining method on the rock mass. The

area can be divided into key monitoring areas, sub-key monitoring areas, and normal monitoring areas. In this way, the effective range factor $F_Q$ is used to mark the region division, as shown in Table 1.

**Table 1.** The effective range factor of different monitoring areas.

| Monitoring Area | Key Areas | Sub-Key Areas | Normal Areas |
|---|---|---|---|
| the effective range factor $F_Q$ | 1.5 | 1.2 | 1.0 |

## 4. Optimization of Microseismic Monitoring Network Design Based on Particle Swarm Optimization

### 4.1. Improved Particle Swarm Algorithm

Particle swarm optimization (PSO) is an evolutionary computing technique, which originated from the study of predation behavior of birds and was first proposed by Kenney and Eberhart [26]. In PSO, the solution to each optimization problem is considered as a particle in the search space [27]. All particles have an adaptive value determined by the optimized function, and each particle also has a velocity to determine its flying direction and distance. Then, the particles search the solution space following the current optimal particle. PSO initializes to produce a group of random particles and then finds the optimal solution through iteration. In each iteration, the particle updates itself by tracking two extremes. One is the optimal solution reached by each particle in the search through the ages, which is called individual extremum $P_{best}$. The other is the optimal solution reached by all particles in the whole particle swarm in the search of previous generations, which is called the global extreme value $g_{best}$. The position of the $i^{th}$ particle in the population in n-dimensional space is expressed as $x_i = (x_{i1}, x_{i2}, \cdots, x_{in})$, and its velocity is $v_i = (v_{i1}, v_{i2}, \cdots, v_{in})$. When finding these two extremes, update the speed and position with the following equation:

$$v_i(k+1) = wv_i(k) + c_1 rand_1(P_{best} - x_i(k)) + c_2 rand_2(g_{best} - x_i(k))$$

$$x_i(k+1) = x_i(k) + v_i(k+1) \tag{6}$$

where $k$ is the iteration number; $c_1$ and $c_2$ are learning factors, which are usually taken between (0, 2); $rand_1$ and $rand_2$ are random Numbers between (0, 1); and $w$ is the momentum coefficient, whose value can change with algorithm iteration [28].

The traditional particle swarm optimization (PSO) algorithm has defects such as a slow iterative speed and easily falling into local extremums. Bacterial foraging optimization (BFO) is a swarm intelligence algorithm inspired from the foraging behavior of the *E. coli* bacteria. The BFO is based on three basic processes: chemotaxis, reproduction, and elimination-dispersal [29]. The bacterial foraging algorithm was introduced to the particle swarm optimization (PSO) algorithm, which will be added to the chemotaxis behavior and the elimination-dispersal behavior, such as in Equations (7) and (8), to increase the randomness of particle movement and modify the particle fitness value [30]. When calculating fitness value, the chemotaxis behavior should be increased to improve the searching speed of particles. When the particle is trapped in the local optimum, the elimination-dispersal behavior is increased to rapidly change the direction and carry out the random walk:

$$Fv_i^{cc}(k) = \sum_{i=1}^{l} [-d_{attr} \exp\left(-w_{attr} \sum_{m=1}^{p} (x_m(k) - x_i^m(k))\right)^2 + \sum_{i=1}^{l} [-h_{rep} \exp\left(-w_{rep} \sum_{m=1}^{p} (x_m(k) - x_i^m(k))\right)^2]$$

$$Fv_i(k) = Fv_i(k) + Fv_i^{cc}(k) \tag{7}$$

$$x_i(k) = x_i(k) + c_i(k)\frac{\Delta(k)}{\sqrt{\Delta^T(k)\Delta(k)}} \quad \text{(trapped in the local optimum)} \tag{8}$$

where $C_i(k)$ is the forward moving step of particle $i$ in $k^{th}$ iteration, and $\Delta(k)$ is a unit vector in the random direction at $k^{th}$ iteration and takes the value $[-1, 1]$. $Fv_i^{cc}(k)$ is the modified increment value of the fitness function of particle $i$ in the $k^{th}$ iteration, which reflects the sum of the gravitational and repulsive forces generated by the whole particle swarm at the position of particle $i$. $Fv_i(k)$ is the fitness function value of particle $i$ in $k^{th}$ iteration; $d_{attr}$ is the depth of gravity, $w_{attr}$ is the width of gravity, $h_{rep}$ is the height of repulsion, $w_{rep}$ is the width of repulsion, $x_{mi}$ is $m^{th}$ component of particle $i$, and $x_m$ is the $m^{th}$ component of other particles in the whole particle population.

### 4.2. Objective Function Construction

To sum up, the microseismic events are locally concentrated and distributed discretely in the whole area due to the existence of faults and broken zones and the diversity of mining techniques. The distribution characteristics are directly related to mine production and safety. It is necessary to increase the number of sensors in key areas. While it is only necessary to monitor large-scale events such as mine earthquakes, a few sensors can be arranged in some areas. Therefore, factors such as the importance of the monitoring area and the possibility of microseismic occurrence need to be considered.

According to the above analysis, the microseismic probability factor $F_e$, the microseismic importance factor $F_Q$, and the effective range factor $F_V$ are introduced to establish the objective function of the optimal network layout scheme. The microseismic monitoring network optimization problem can be transformed into the $D$-value optimization problem shown in Equation (9):

$$obj = \min\left[P_V\left(\sum_{j=1}^{n_Q}\sum_{i=1}^{ne} P_{Qj}(h_i)P_e(h_i)\lambda x_0(h_i)\lambda y_0(h_i)\lambda z_0(h_i)\lambda t_0(h_i)\right)\right] \qquad (9)$$

In the equation, $F_V$ is the effective range factor. $F_{Qj}$ is the importance factor in the $j^{th}$ area of the microseismic monitoring space. $F_e$ is the microseismic probability factor. $n_Q$ is the number of areas divided into the monitoring space, and other parameters are as in Equation (3).

### 4.3. Evaluation Indicators of Network Design

When designing a network of microseismic monitoring stations in an actual mine, the standard error map of the seismic event parameters H$\{t_0, x_0, y_0, z_0\}$ corresponding to the station layout scheme obtained by the above method is expressed by S. J. Gibowicz and A. Kijko [31]. The source error is shown in Equation (10):

$$\begin{aligned} \sigma_{xy} &= \left[(C_x)_{22}(C_x)_{33} - [(C_x)_{23}]^2\right]^{\frac{1}{4}} \\ \sigma_z &= [(C_x)_{44}]^{1/2} \\ \sigma_{xyz} &= \left[(\sigma_{xy})^2 + (\sigma_z)^2\right]^{1/2} \end{aligned} \qquad (10)$$

In the equation, $(C_x)_{ij}$ is the matrix $C_x$ of the element $(i, j)$. $\sigma_{xy}$ is the epicenter error. $\sigma_z$ is the focal depth error, and $\sigma_{xyz}$ is the focal spatial error.

Whether a microseismic monitoring system with a certain monitoring station network configuration can detect earthquake events with magnitude $M_L$ at H$_i$ constitutes the concept of monitoring system's sensitivity [32]. The expected standard deviation graph drawn by Equation (10) is a function of the event magnitude; that is, the equation represents the source location standard error whose magnitude is $M_L$ and whose source coordinates are H$_i$. In the proposed monitoring area, the event magnitude $M_L$ can be related to its measurable distance $r$. The distance $r$ from the point H$_i$ to the monitoring station can be calculated, and then the distance can be converted into an earthquake magnitude, and a sensitivity contour map can be drawn. All stations within this distance $r$ are used to calculate the expected error of the source [14]. The ideal microseismic station network layout must have good

sensitivity and positioning error. Therefore, we combine the sensitivity contour map and the positioning error map to evaluate the network array.

### 4.4. Design Process of Microseismic Monitoring Network Based on IPSO

The microseismic network optimization layout system consists of two modules, namely, the station optimal scheme selection of Module I and the station scheme evaluation of Module II. According to the algorithm flow in Figure 1, in Module I, the improved particle swarm algorithm proposed (IPSO) in this paper is used for microseismic network layout optimization following the principle of not only taking care of the current mining area but also considering the mining scheme in a certain period in the future, according to the delineation of the dangerous area of the mine and geological factors, etc. In Module II, the positioning error and sensitivity of the current network are evaluated. The specific process is shown in Figure 1.

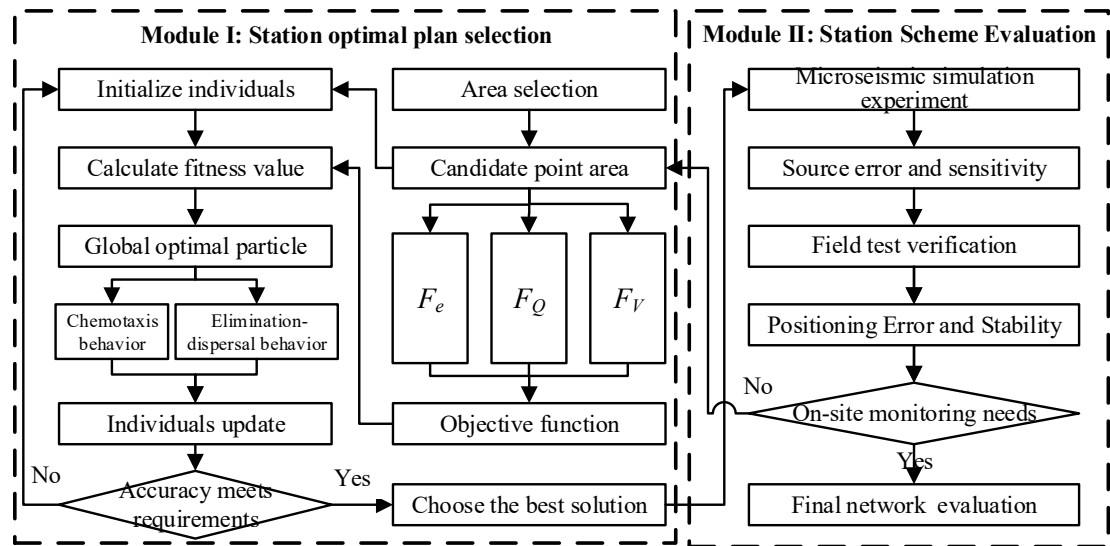

**Figure 1.** Flow chart of the microseismic network layout optimization.

(1) Module I: Station optimal scheme selection:
- ① According to the mining technical factors and geological factors related to the risk of rock burst on site, select the microseismic monitoring area, divide the structural area into a geographic grid, and determine the range of candidate points for the station.
- ② According to the on-site investigation or numerical analysis method, determine the microseismic probability factor $F_e$, the microseismic importance factor $F_Q$, and the effective range factor $F_V$ to form a set of station candidate points.
- ③ According to the *D*-value optimization theory, establish the station optimization objective function of Equation (9).
- ④ Use the improved particle swarm algorithm (IPSO) proposed in this paper to select the optimal station scheme.

(2) Module II: Station Scheme Evaluation:
- ① According to Equation (10), the sensitivity contour map and the positioning error map of the current network are calculated to preliminarily evaluate the feasibility of the network layout.
- ② Carry out the field test of percussion or blasting in the mine to obtain the positioning accuracy and stability of the current network layout scheme and further evaluate the pros and cons of the network layout.
- ③ According to the above results, judge whether the performance of the station network can meet the needs of on-site microseismic monitoring.

## 5. Theory and Field Test Verification

### 5.1. Theoretical Experimental Research

To verify the superiority of IPSO in solving the optimal arrangement of the microseismic monitoring network, a hypothetical artificial network was studied. For the mining working face, we assume that the X direction interval of the monitoring model is [0 m, 100 m], the Y direction interval is [0 m, 50 m], and the elevation is 200 m. The key monitoring area range is [200 m, 500 m] in the X direction, [0 m, 200 m] in the Y direction, and the elevation is 200 m. The grid spacing is 100 m × 50 m, and the sensor arrangement elevation is 0 m. The probability of shaking at all grid nodes is 1.0. The distribution of candidate points for all 18 stations is shown in Figure 2. The key monitoring area is the yellow area in Figure 2. According to the provisions of Table 1, $F_Q$ of the key monitoring area is 1.5, and the $F_Q$ of other monitoring areas is 1.0.

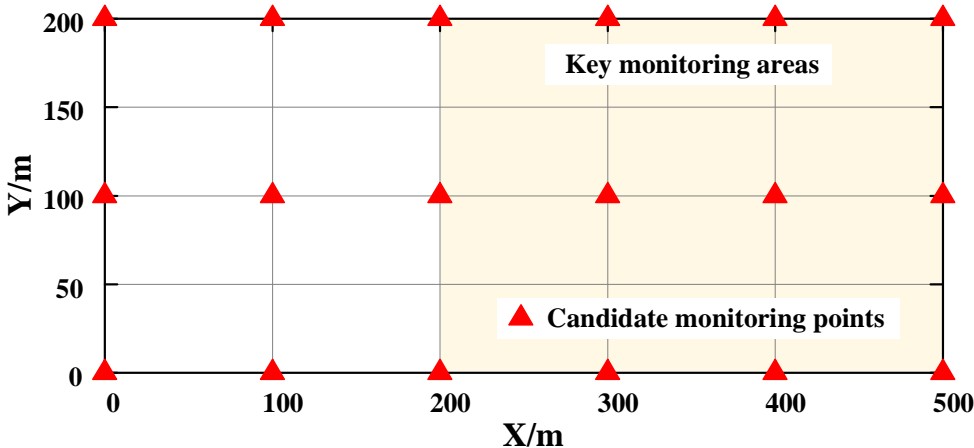

**Figure 2.** Distributions of candidate microseismic monitoring points.

To simplify the solution process, it is assumed that all vibrations in the monitoring area can be monitored, and the expected value of the P-wave velocity is 3200 m/s. The test is divided into three schemes. Scheme I selects four candidate points from the set of the optimal network layout scheme. Scheme II selects six candidate points from the set. Scheme III selects eight candidate points from the candidate point set. The three schemes are solved by using IPSO, the simulated annealing algorithm (SA), and the exhaustive algorithm (EA), respectively. IPSO parameters are set as follows: the learning factor $c_1 = c_2 = 2$, the group size $N_{pop} = 100$, $w_0 = 1$, $C = 10$, $d_{attr} = w_{attr} = 0.05$, $h_{rep} = w_{rep} = 0.05$, the end condition $\varepsilon_0 = 0.001$, and the search times $E_t = 50$. The parameters of SA are set as follows: the initial temperature $t_0 = 97$, the end temperature $t_f = 89.9$, and the temperature drop ratio $\lambda = 0.99$.

Table 2 lists the results of different network layout schemes. It can be seen that the candidate points layout schemes show multiple increases with the increase in the number of sensors. The model solution results are shown in Table 3 and Figures 3 and 4. From the results of EA, it can be seen that, except for scheme I, which has multiple solutions, both schemes II and III have unique solutions. Since the monitoring area is a cube, it can be seen that there is symmetry among the multiple solutions through rotation. Although SA and IPSO cannot obtain all the optimal solutions, the global optimal solutions are found for the three schemes. However, more importantly, the two optimization algorithms do not significantly increase the computation time as the number of combinations increases. After the number of probes increased to eight, the time-consumption of EA increased to 261.88 s, while SA needed 37.4 s and IPSO needed 27.06 s to obtain the optimal network layout. Therefore, EA and IPSO are global in the optimization of the network optimization problem, and they are also efficient in time. As can be seen from Figure 5, for scheme I, IPSO found the optimal solution when it evolved to the 10th generation, and SA obtained the optimal solution in the 38th generation. In scheme II and scheme III, IPSO finds the optimal solution in the 22nd and the 6th generations, respectively, which are better than

the 29th and the 31st generations of SA. It can be seen that the time-consumption of IPSO should be shorter.

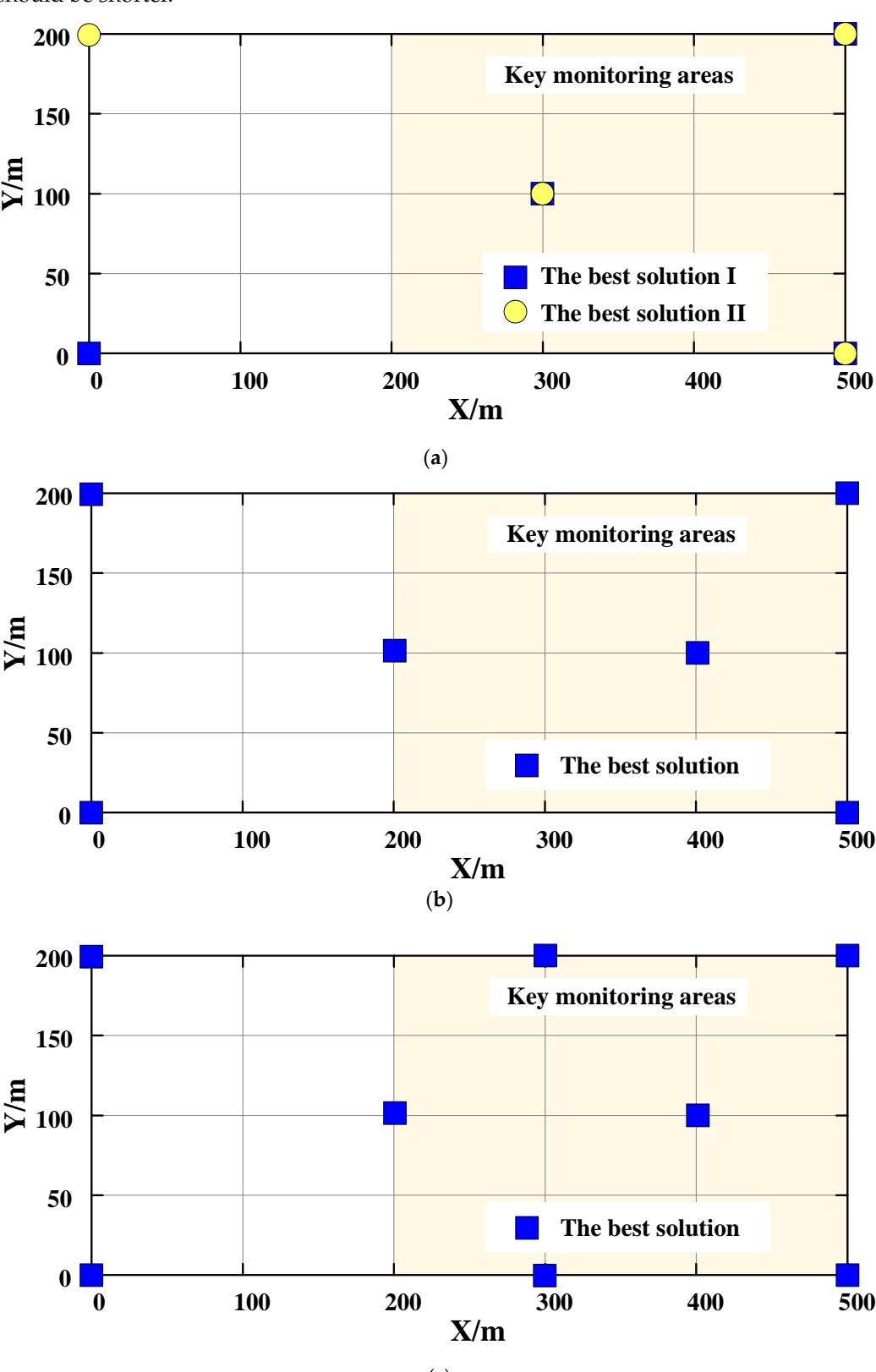

**Figure 3.** Network configuration results solved by EA. (**a**) Solution Results of Scheme II. (**b**) Solution Results of Scheme II. (**c**) Solution Results of Scheme III.

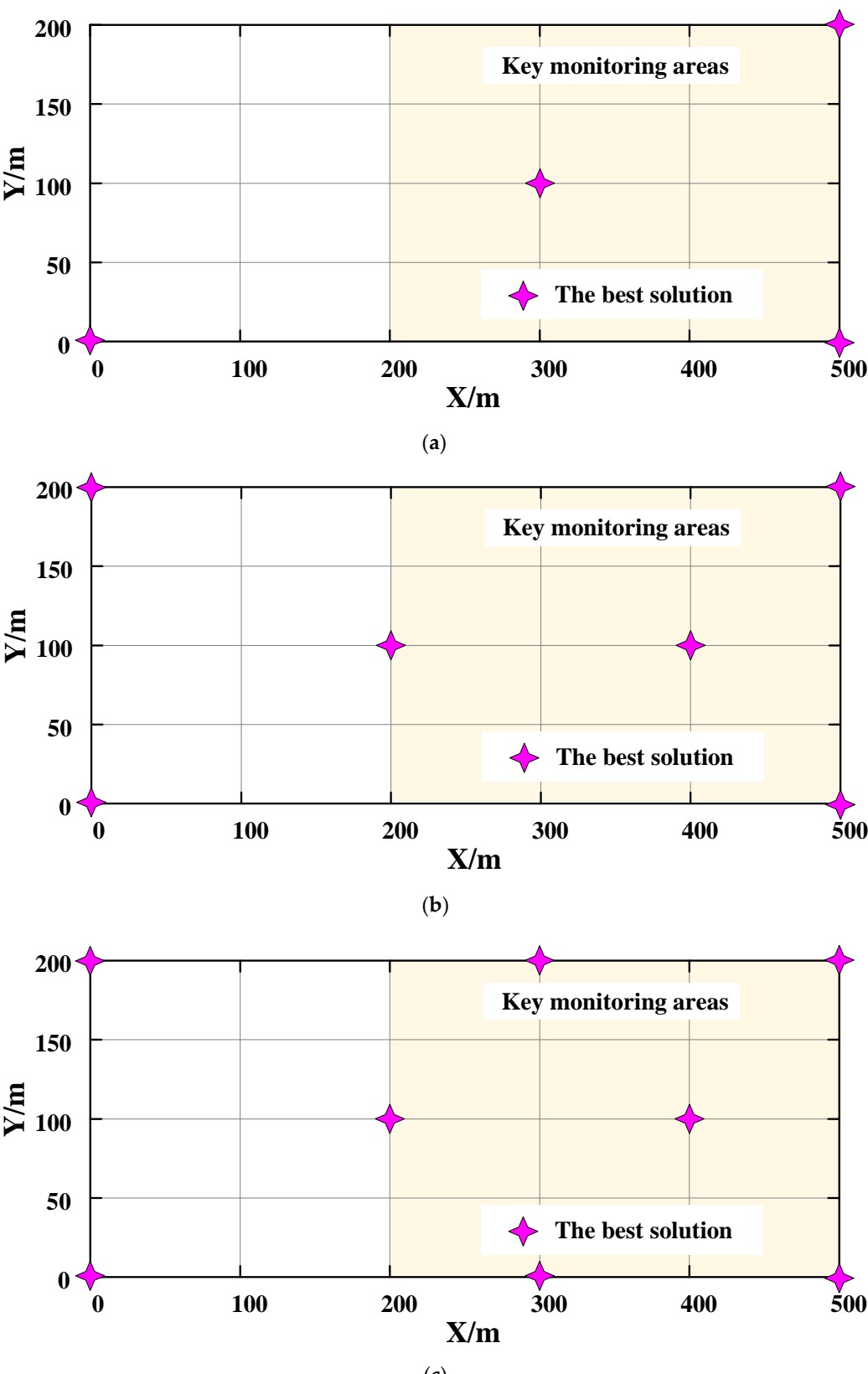

**Figure 4.** Network configuration results solved by IPSO and SA. (**a**) Solution Results of Scheme I. (**b**) Solution Results of Scheme II. (**c**) Solution Results of Scheme III.

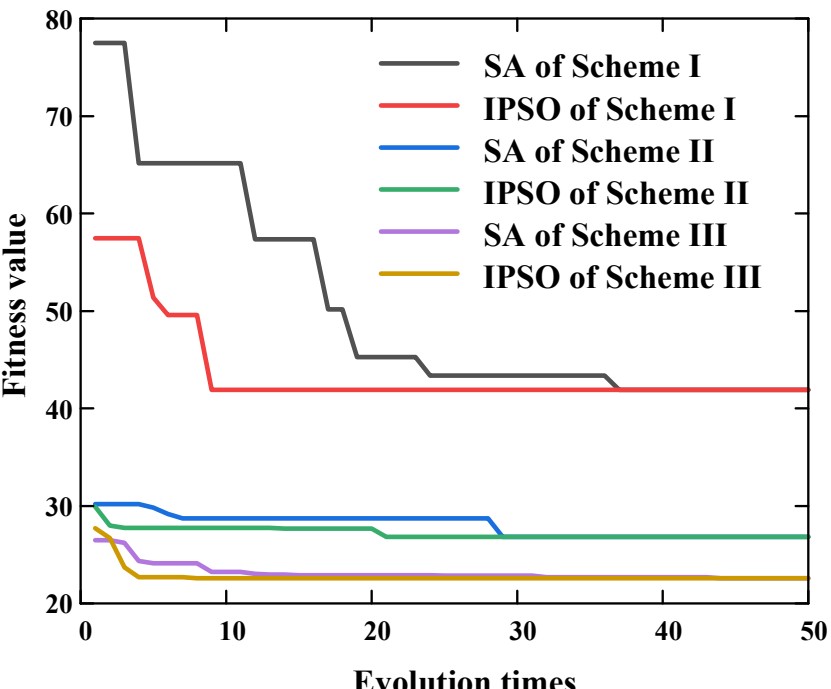

**Figure 5.** Comparison of optimization curves between IPSO and SA.

**Table 2.** Comparison of the results of different network layout schemes.

| Schemes | Monitoring Points | Sensors | Combinations | Results |
|---|---|---|---|---|
| Scheme I | 18 | 4 | 3060 | Figure 3a |
| Scheme II | 18 | 6 | 18564 | Figure 3b |
| Scheme III | 18 | 8 | 43758 | Figure 3c |

**Table 3.** Comparison of optimization results of different algorithms.

| Schemes | Running time | | | Solutions | |
|---|---|---|---|---|---|
| | EA | SA | IPSO | EA | EA and IPSO |
| Scheme I | 48.01 | 17.74 | 15.63 | Figure 3a | Figure 4a |
| Scheme II | 144.32 | 26.65 | 22.55 | Figure 3b | Figure 4b |
| Scheme III | 261.88 | 37.4 | 27.06 | Figure 3c | Figure 4c |

The source error and sensitivity of different sensor arrangement schemes are calculated by Equation (10). The evaluation results are shown in Figures 6 and 7. It can be seen from Figure 6 that the source error of the six-channel sensor in scheme II is significantly lower than that of the four-channel sensor in scheme I, and the source error of the eight-channel sensor in scheme III is even lower than the previous two, keeping the source error in key areas within 50 m. It can be seen that as the number of sensors increases, the positioning error decreases significantly. The monitorable magnitude of the key monitoring area of Scheme II is −3.5481 in Figure 7, which is better than that of Scheme I, which is −3.1222. However, the sensitivity of scheme III and scheme II is the same. In addition, it can be obtained by calculation that the effective range factor $F_V$ of scheme I is 0.44, the effective range factor $F_V$ of scheme II is 0.75, and the effective range factor $F_V$ of scheme III is 0.87. It can be seen that the arrangement scheme of the eight-channel sensor of scheme III has the best monitoring advantage.

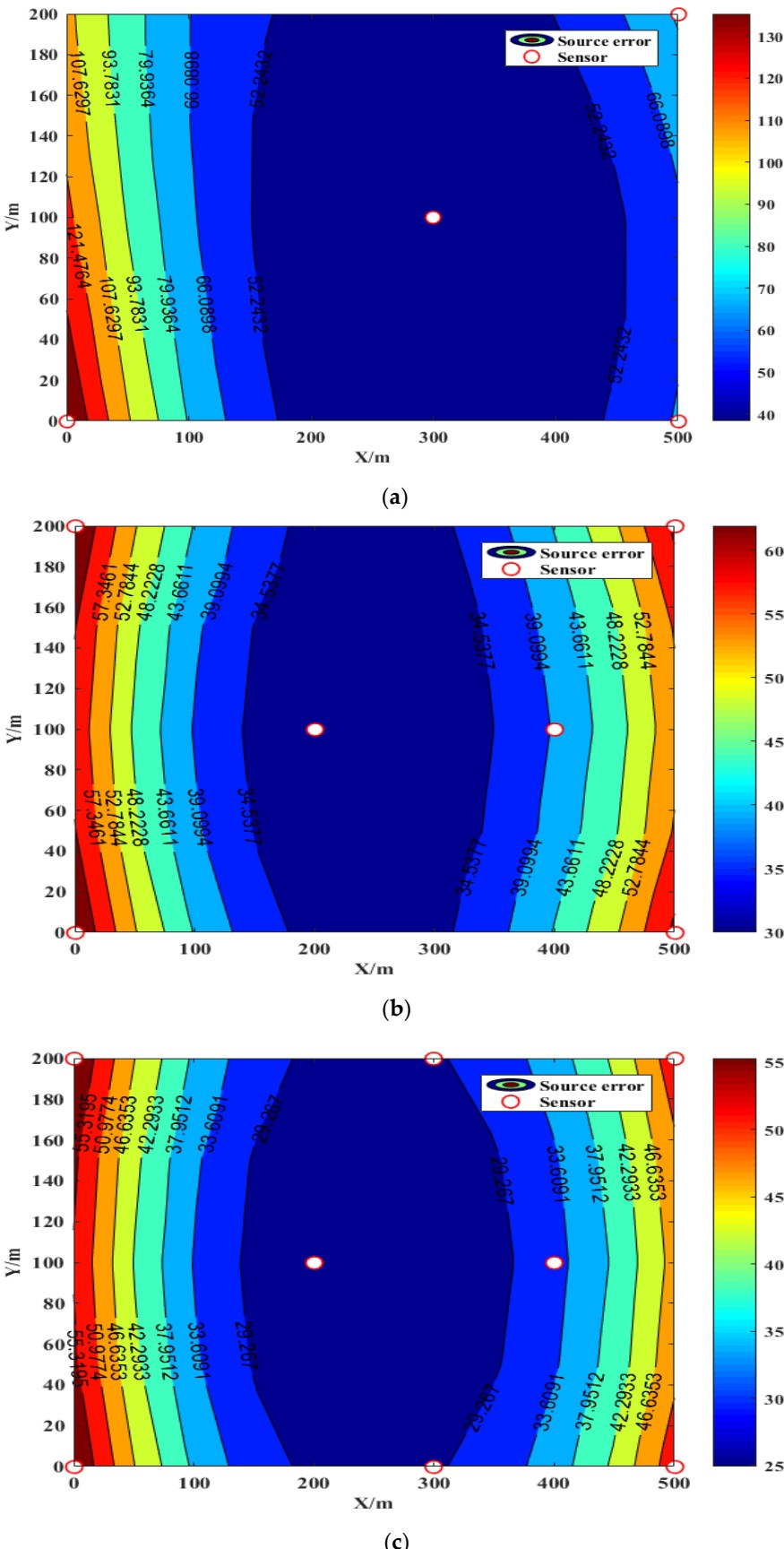

**Figure 6.** Error evaluation of different network layout schemes. (**a**) Positioning error of Scheme I. (**b**) Positioning error of Scheme II. (**c**) Positioning error of Scheme III.

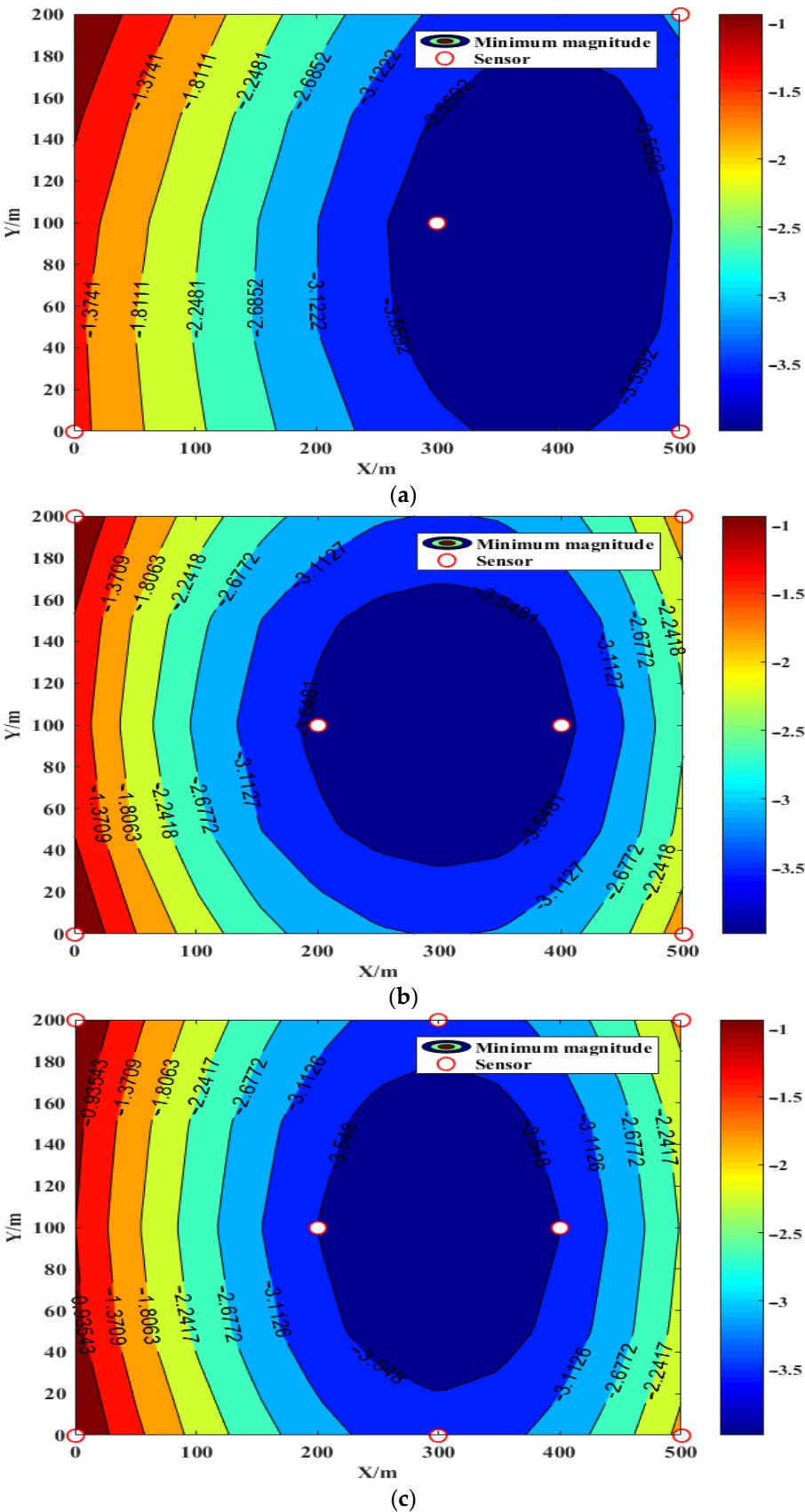

**Figure 7.** Sensitivity evaluation of different network layout schemes. (**a**) Sensitivity of Scheme I. (**b**) Sensitivity of Scheme II. (**c**) Sensitivity Result of Scheme III.

### 5.2. Field Application Studies

5.2.1. Determination of Monitoring Area and Candidate Point Scheme

Xiashijie coal mine is located in Yaoqu town, Tongchuan City, Shaanxi Province, China, and it is 54 km away from Tongchuan city, as shown in Figure 8. The minefield is 4 km long, with a stope width of 3.3 km and a coal-bearing area of 13.2 square kilometers. The mine design is two levels, which include the 'Cross Double U' roadway layout of the mining face and the adit-inclined shaft stage mining. Longwall mining and fully mechanized top coal caving are adopted, and the total collapse method manages the roof. The general structure of the mine is an undulatory monocline with a north dip, and the deep part is a syncline of Xinmin village. The fault structure is not developed and is simple.

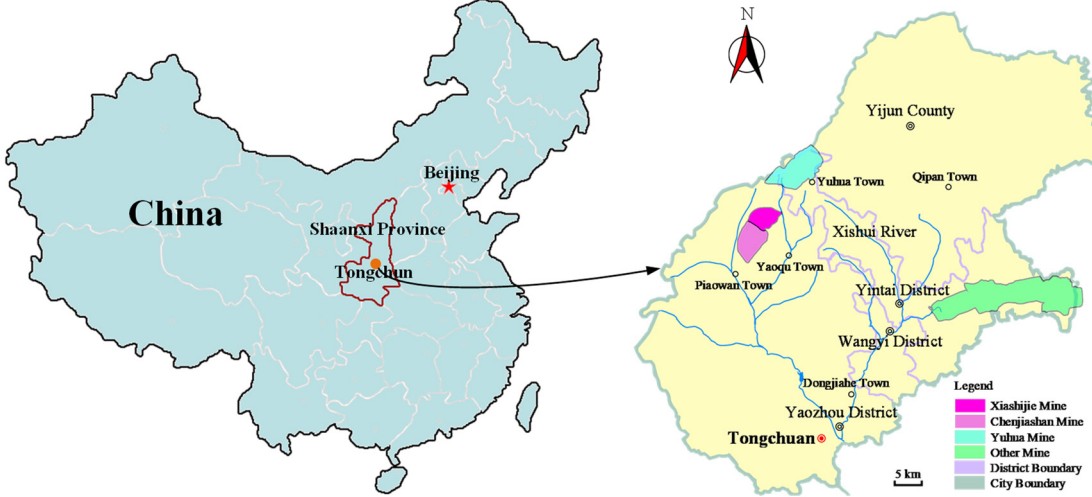

**Figure 8.** Location of Xiashijie coal mine.

In view of the randomness and multiple problems of mine rock-burst, to reveal and predict potential risk areas of the working face in advance, our research group adopted the advanced high-precision microseismic monitoring system produced by the Canadian Engineering Seismology Group (ESG) in October 2019 to carry out real-time monitoring, positioning, and analysis of coal and rock fractures in the affected mining area of the working face 2305. The system consists of an underground signal acquisition system, a ground data processing system, and a remote system. The underground signal acquisition system includes microseismic sensors and the Paladin downhole digital signal acquisition system. The remote system contains a big data processing system and 3D visualization software based on remote network transmission developed by Mechsoft (Dalian) Co., Ltd. The microseismic sensors use seismometers with a response frequency range of 15–1000 Hz and a sensitivity of 43.3 v·s/m.

The mine was in the mining stage. The mining position is shown in Figure 9. There is a syncline at a distance of 300 m from the incision. The working face about 50 m before and after the syncline is used as the microseismic risk area, and the probability of mine earthquakes increases by 20%. The mining face within the mining stop line is taken as the key monitoring area, and the microseismic importance factor $F_Q$ of the key monitoring area is 1.5. According to the general principles of microseismic sensor arrangement, on-site mining technology, and geological conditions, 82 candidate points are selected in the installation area with a grid interval of 50 m according to the size of the layout area, since the installation positions are mainly arranged in the haulage roadway and the air return roadway. Using 20 sensors, the improved particle swarm algorithm (IPSO) proposed in this paper is used to optimize the design of the network optimization.

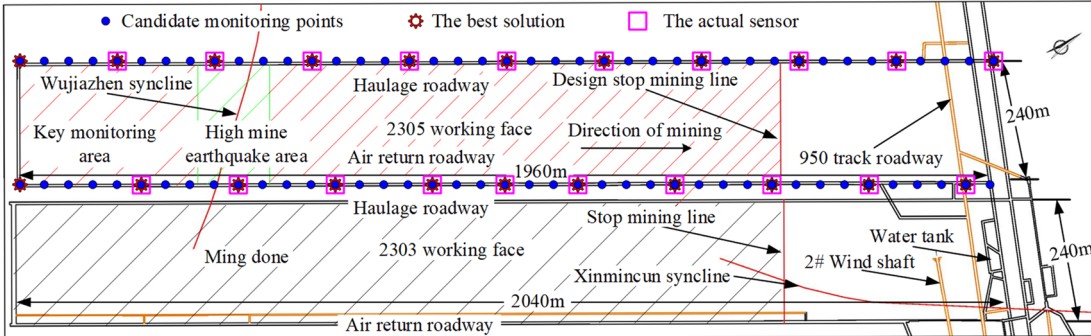

**Figure 9.** Layout scheme of on-site microseismic network.

5.2.2. Solving the Optimal Solution

Due to a large number of candidate points, if the exhaustive method is used, there are a total of $3.5352 \times 10^{18}$ solutions, which is not feasible in terms of time. However, using IPSO to solve the problem, it is found that the algorithm has a fast convergence speed (see Figure 10), and the running time is 244.86 s, the individual population has converged to the optimal solution when it has evolved to the 50th generations. The algorithm parameters are set as in Section 5.1. According to the field test, the expected velocity of the P wave is taken as 4200 m/s. According to the above setting parameters and models, the optimal layout scheme is shown in Figure 9.

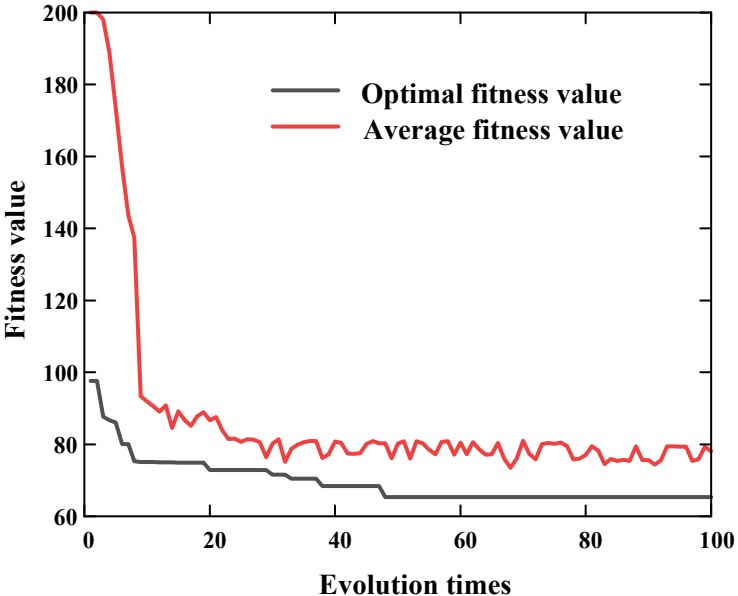

**Figure 10.** Optimization curve of IPSO.

Combined with the on-site installation conditions, the actual installation position of the microseismic sensor is shown in Figure 9, which is dragged 200 m forward along the incision position. On the one hand, this adjustment is beneficial to the recovery of the on-site sensors, and on the other hand, it can effectively monitor the ore body at the working face behind the stopped mining line.

5.2.3. Evaluation of the Positioning Capability

(1) Numerical simulation evaluation

The source location accuracy and sensitivity are calculated for the proposed site layout scheme. According to the test results of the acoustic characteristics of the mine, the P-wave velocity is 4200 m/s, and the P-wave arrival error is 1.5 ms. Drawing the positioning accuracy map, take the magnitude $M_L = 1$. Drawing the sensitivity map, the effective

number of digits is 5, and the minimum peak particle velocity that the microseismic sensor can resolve is 0.09 mm/s. Draw the expected standard error map and the contour map at different depths, as shown in Figures 11 and 12.

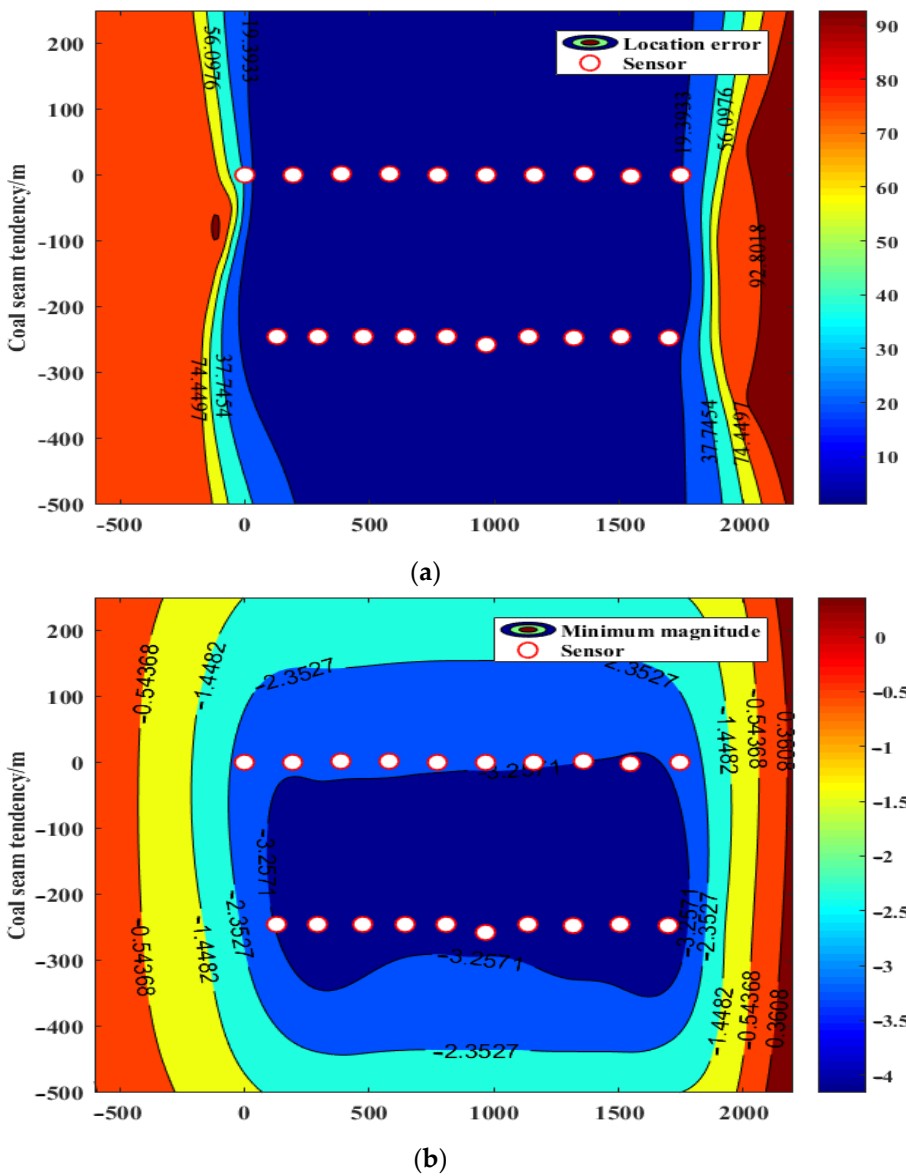

**Figure 11.** Evaluation of sensor arrangement on the coal seam floor (920 m). (**a**) Positioning errors. (**b**) Sensitivity.

Figures 11 and 12 show the positioning error and sensitivity distribution of the bottom plate (920 m) and the top plate (970 m) of the working surface 2305, respectively. Different colors are used to represent the positioning error and sensitivity. The numerical units on the positioning error color scale and the sensitivity color scale are m and Richter scale values, respectively. It can be seen that for the coal seam roof and floor of the working face 2305, the source positioning errors of the key monitoring areas are all within the range of less than 20 m, indicating that the working face back to the mining body and its surrounding rocks are basically in the area with high source positioning accuracy. In addition, the source positioning accuracy is still in the range of less than 20 m within the range of not less than 200 m along the working face. The source positioning accuracy decays rapidly as it moves away from the sensor array along the working face. Therefore, from the perspective of source location accuracy, the sensor arrangement scheme not only satisfies the requirements of location accuracy but also makes the microseismic monitoring system economical.

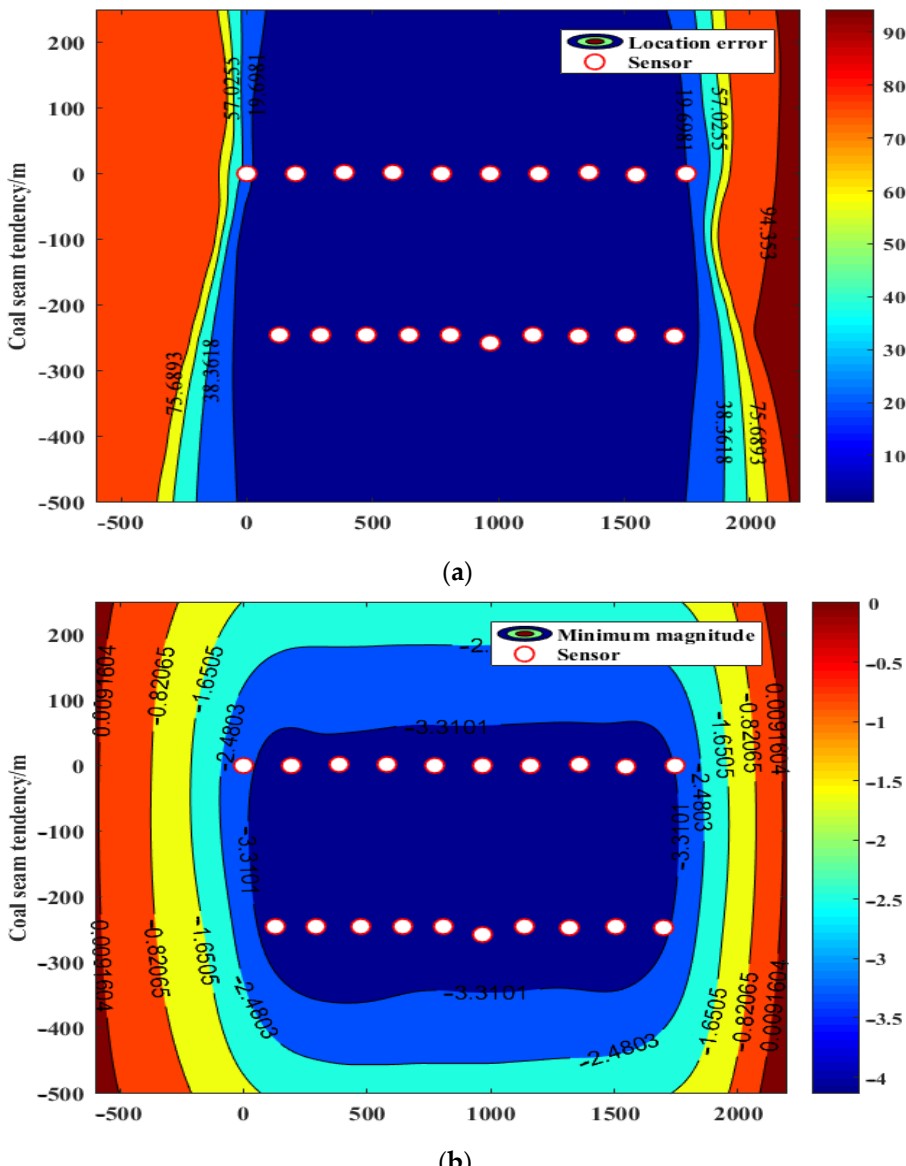

**Figure 12.** Evaluation of sensor arrangement on the coal seam roof (970 m). (**a**) Positioning errors. (**b**) Sensitivity.

It can be seen from Figures 11 and 12 that, regardless of the mine roof or floor, the minimum magnitude that can be measured in the ore body and its surrounding rock of the key monitoring area is −3.26, and the local position of the floor can reach −3.31. In the area around the working face, the minimum measurable magnitude of the top plate is −2.35, and that of the bottom plate is −2.48. Therefore, the sensor arrangement scheme has sufficient system sensitivity.

(2)　Field test evaluation

To further evaluate the rationality of the network layout, taking the three blasting tests of the mine stope on 8 November 2019 as examples, the positioning calculation was carried out according to the coordinates of the geophone and the corresponding arrival time. Compared with the measured blasting position, the positioning coordinates are as shown in Table 4. Using the simplex localization method in the microseismic system, the blasting event was effectively located. The test results are shown in Table 5.

**Table 4.** Blasting point positioning coordinates.

| Serial Number | Blasting Coordinates | | | Positioning Coordinates | | |
|---|---|---|---|---|---|---|
| | x/m | y/m | z/m | x/m | y/m | z/m |
| 1 | 2710.5 | 3647.4 | 935.5 | 2709 | 3640 | 938.1 |
| 2 | 2220.6 | 3302.8 | 938.7 | 2215 | 3310 | 933.4 |
| 3 | 2065.9 | 3201.3 | 925.8 | 2061 | 3210 | 932.9 |

**Table 5.** Blasting test results.

| Serial Number | Time | Errors | | | Absolute Error/m |
|---|---|---|---|---|---|
| | | x/m | y/m | z/m | |
| 1 | 15:35 | 1.5 | 7.4 | 2.6 | 8 |
| 2 | 15:54 | 5.6 | 7.2 | 5.3 | 10.6 |
| 3 | 17:55 | 4.9 | 8.7 | 7.1 | 12.2 |
| Max | - | 5.6 | 8.7 | 7.1 | 12.2 |
| Mean | - | 4.0 | 7.8 | 5.0 | 10.3 |

From the comparison in Table 5, it can be seen that the maximum positioning errors of the three coordinate directions are 5.6 m, 8.7 m, and 7.1 m, respectively. The average positioning errors are 4.0 m, 7.8 m, and 5.0 m, respectively, the maximum spatial positioning error is 12.2 m, and the average is 10.3 m. Therefore, the positioning accuracy can meet the needs of engineering monitoring. From the coverage rate of the monitoring area, the effective range factor $F_V$ of the coal seam floor is 0.7831, while the effective range factor $F_V$ of the coal seam roof is 0.7880. The above research shows that the layout of the microseismic station network in Xiashijie Coal Mine is reasonable and meets the needs of the mine's microseismic monitoring.

## 6. Conclusions

(1) With the *D*-value optimization design theory and the good global optimization ability of IPSO, the microseismic probability factor $F_e$, the microseismic importance factor $F_Q$, and the effective range factor $F_V$ are introduced to establish the objective function of the optimal network layout scheme. The monitoring sensitivity and positioning error are used as evaluation criteria, and an optimization and evaluation system for the optimal layout of a mine's microseismic network is proposed. The system consists of the station optimal scheme selection of Module I and the station scheme evaluation of Module II.

(2) Using numerical simulation experiments, the improved particle swarm algorithm (IPSO) proposed provides a feasible method for the optimal design of the microseismic network. When there are a large number of candidate points, the method in this paper can quickly and cost-effectively complete the optimal network layout. In addition, the superiority of the optimal scheme is verified by the evaluation method.

(3) Taking Xiashijie Coal Mine in Tongchuan, Shaanxi Province, as an example, IPSO took 244.86 s to optimize the arrangement of sensor candidate points in the key monitoring area of the mining area, according to the on-site geological conditions. The research results show that the algorithm can quickly find the optimal solution, the errors in key monitoring areas are all within 20 m, and the minimum monitoring magnitude is −3.26. The blasting test analysis shows that with the network layout scheme proposed in this study, the maximum error in the comprehensive location of the source is 12.2 m, and the average is 10.3 m. The positioning accuracy can meet the needs of engineering monitoring, and related research can provide a reference for the microseismic layout of similar projects.

**Author Contributions:** Conceptualization, K.W. and C.T.; methodology, K.W., C.T. and K.M. software, K.W.; validation, K.W. and T.M.; formal analysis, K.M. and K.W.; investigation, K.W.; data curation, K.W. and K.M. writing—original draft preparation, K.W.; writing—review and editing, K.M. and T.M.; visualization, K.W.; supervision, T.M.; project administration, C.T.; funding acquisition, C.T. All authors have read and agreed to the published version of the manuscript.

**Funding:** This research was funded by the National Natural Science Foundation of China, grant number 41941018 and 51627804.

**Institutional Review Board Statement:** Not applicable.

**Informed Consent Statement:** Informed consent was obtained from all subjects involved in the study.

**Data Availability Statement:** The data presented in this study are available on request from the corresponding author. The data are not publicly available due to privacy concerns.

**Acknowledgments:** The technical staff of Tongchuan mining Co., LTD., are appreciated for their assistance. Wang Xintang and Li Qiang are appreciated for their support with on-site data.

**Conflicts of Interest:** The authors declare no conflict of interest.

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
