# Peer review of "Research on the Design of Coal Mine Microseismic Monitoring Network Based on Improved Particle Swarm Optimization"

_applsci, doi:10.3390/app12178439_

Round 1
Reviewer 1 Report
The paper describes a methodology for an optimised design of a monitoring network for microseismic events induced by coal mining. The topic is of interest and suitable for Applied Sciences Journal. However, some improvements must be provided. Please, see my observations in the following.
1) The abstract is too long (348 words). According to the Journal's Instructions for Authors it should be maximum 200 words. Moreover, it is sometimes confusing being the real novelty of the work not very clear.
2) Lines 13-15: “The bacterial foraging algorithm is introduced into the particle swarm algorithm (PSO), and the directional behavior and the focusing behavior are added to the swimming of the particles, and an improved particle swarm algorithm (IPSO) is proposed.”. After reading the introduction, it seems that this algorithm is not introduced in this paper but just adopted. Please clarify.
3) Lines 54-56: The authors should consider better explaining what the D and C values are because the readers may not be experts on optimal design theory.
4) Introduction: Please clarify better the novelty of your paper. Do you have introduced the PQ and PV factors in the IPSO algorithm as stated in Line 105-106?
5) Line 118: Again, as Comment 3. It is not clear what is the D-value theory. From your references, it seems more appropriate to refers to this method as D-optimal design theory.
6) Eq. (1). Why do you add three times (x0-xi)2? What does it mean? Is it a mistake?
7) Line 139-141: It is not clear what the authors mean with “hypocenter points calculated…”. How do you calculate them?
8) Line 152: You write about geophones, even if you used seismometers in your application (Line 393). It could be useful to explain what kind of measurement instruments can be used.
9) Line 175: What do you mean with monitoring range?
10) Line 196: How do you evaluate the effective range factor?
11) Line 213 and 215: the definition of gbest is repeated.
12) Eq. (6): Consider dividing this equation into (6a) and (6b)
13) Lines 223-224: Equations (12) and (13) are missing.
14) Line 259-260. Fix all the subscripts.
15) Line 277: “the improved particle algorithm proposed in this paper”. Please, clarify if the algorithm already existed or if you have written it.
16) Lines 311-312. From Figure 2 it seems that the interval in Y direction is [0 200], not [0 500] how you wrote. Moreover, again in Y direction, the grid spacing is 50 m (You wrote “The grid spacing is 100”).
17) Lines 327-328: It is the first time I read about temperature in this paper. It is never mentioned before. The same symbol (capital letter T) is used in Eq.(1) referring to time. Maybe, you can adopt a lowercase “t” for time.
18) Line 156: It is preferable to refer to Equation as “Eq. (10)” not “formula”.
Author Response
Response Letter- Reviewer 1
First of all, the authors highly appreciate the editors and reviewers’ comments on the manuscript. We carefully reviewed the comments and suggestions and have incorporated major revisions into the manuscript. Hopefully, the revised version could satisfy the reviewers. We also welcome further comments and suggestions if they may have. Listed below are the authors’ point-by-point responses to the reviewers’ comments. The revisions are highlighted red in the revised manuscript.
The paper describes a methodology for an optimised design of a monitoring network for microseismic events induced by coal mining. The topic is of interest and suitable for Applied Sciences Journal. However, some improvements must be provided. Please, see my observations in the following.
- The abstract is too long (348 words). According to the Journal's Instructions for Authors it should be maximum 200 words. Moreover, it is sometimes confusing being the real novelty of the work not very clear.
Response 1:
Thanks a lot for your question suggestion. It should be noted that the innovations of this paper are: (1) In the D-value optimization design theory, the microseismic probability factor Fe, the microseismic importance factor FQ, and the effective range factor FV, enrich the sensor optimization objective function. (2) The bacterial chemotaxis algorithm was introduced into the particle swarm algorithm, which increased the global search ability of the algorithm, and applied the improved particle swarm algorithm to the optimal arrangement of sensors. (3) A comprehensive system of sensor optimization and evaluation is proposed, which can comprehensively evaluate the positioning error and sensitivity of the sensor arrangement. The abstracts at Lines 10 - 22 in the text have been simplified, see below.
With the increasingly severe mining environment, microseismic monitoring is more and more widely used. To ensure the validity of microseismic monitoring results and improve the reliability of early warning and forecasting, microseismic network optimization and evaluation system should be established. The bacterial foraging algorithm is introduced into the particle swarm algorithm (PSO), and the directional behavior and the focusing behavior are added to the swimming of the particles, and an improved particle swarm algorithm (IPSO) is proposed. With the D-value optimization design theory and the good global optimization ability of IPSO, the microseismic probability factor Pe, the microseismic importance factor PQ, and the effective range factor PV are introduced to establish the objective function of the optimal network layout. Monitoring sensitivity and positioning error are used as evaluation criteria, and an optimization and evaluation system for the microseismic network is proposed. The system consists of the station optimal scheme selection of Module I and the station scheme evaluation of Module II. It is found that IPSO not only successfully obtains the best candidate points of the network, but also has the fastest optimization speed, through numerical simulation experiments. The overall efficiency of IPSO for network optimization is better than that of the simulated annealing (SA) algorithm and the exhaustive algorithm (EA). We can see that increasing the number of sensors will be more advantageous from the evaluation indicators of the network layout. Combined with the system application at Xiashijie Coal Mine in Tongchuan City, Shaanxi Province, the method has successfully optimized the layout of the 20-channel network, and the positioning error of the key monitoring areas of the top and bottom of the coal seam is controlled within 20 m, and the minimum measurable magnitude can reach -3.26. Finally, it is verified by the blasting test that the maximum spatial positioning accuracy of the site is within 12.2 m, which accurately evaluates the positioning capability of the site network. The evaluation indicators all meet the needs of on-site microseismic monitoring. The relevant research can provide a reference for the layout of the microseismic monitoring network for similar projects.
is revised to:
The quality of the mine microseismic network layout directly affects the location accuracy of the microseismic. Introducing the microseismic probability factor Fe, the microseismic importance factor FQ, and the effective range factor FV, an improved particle swarm algorithm with bacterial foraging algorithm is proposed to optimize the mine microseismic network layout and evaluation system based on the D-value optimization design theory. Through numerical simulation experiments, it is found that the system has the advantages of fast optimization speed and good network layout effect. Combined with the system application at Xiashijie Coal Mine in Tongchuan City, Shaanxi Province, the method in this paper successfully optimizes the layout of the 20-channel network, ensuring that the positioning error of key monitoring areas is controlled within 20 m, and the minimum measurable magnitude can reach -3.26. Finally, it is verified by blasting tests that the maximum spatial positioning accuracy of the site is within 12.2 m, and the positioning capability of the site network is more accurately evaluated. The relevant research can provide a reference for the layout of the microseismic monitoring network for similar projects.
- Lines 13-15: “The bacterial foraging algorithm is introduced into the particle swarm algorithm (PSO), and the directional behavior and the focusing behavior are added to the swimming of the particles, and an improved particle swarm algorithm (IPSO) is proposed.”. After reading the introduction, it seems that this algorithm is not introduced in this paper but just adopted. Please clarify.
Response 2:
Thank you very much for your comments. The optimization algorithm in this article adopts the improved particle swarm algorithm proposed by the author of this article in 2021. See the reference [30] in the main text. Lines 212 - 218 in the text has added a brief description of the foraging bacteria algorithm, as shown below.
The bacterial foraging algorithm [29] was introduced to the particle swarm optimization (PSO) algorithm, which will be added to the focused behavior and the tropism behavior, namely such as equations (12) and (13) to increase the randomness of particle movement and modify particle fitness value [30].
is revised to:
Bacterial foraging optimization (BFO) is a swarm intelligence algorithm inspired from forging behavior of the E. coli bacteria. The BFO is based on three basic processes; chemotaxis, reproduction, and elimination-dispersal [29]. The bacterial foraging algorithm was introduced to the particle swarm optimization (PSO) algorithm, which will be added to the chemotaxis behavior and the elimination-dispersal behavior, namely such as equations (7) and (8) to increase the randomness of particle movement and modify particle fitness value [30].
- Lines 54-56: The authors should consider better explaining what the D and C values are because the readers may not be experts on optimal design theory.
Response 3:
Thanks a lot for your question. The D-value represents the Determinant value of the source covariance matrix, so that the microseismic sensor arrangement scheme with the smallest D value is the optimal scheme, which is described in detail in Section 2 “Optimization Theory of the Microseismic Network” in the main text. The C-value represents the Condition number of non-linear travel-time equations with respect to the known source parameter vector, so that the microseismic sensor arrangement with the smallest C value is the optimal solution. Lines 45 - 46 in the text have been modified accordingly. The text is as follows.
For example, Kijko [9-10] and Mendecki et al. [11] proposed a microseismic network evalu-ation method based on the optimal design theory of D and C values.
is revised to:
For example, Kijko [9-10] and Mendecki et al. [11] proposed a microseismic network evaluation method based on the optimal design theory of D and C values (Determinant value of the source covariance matrix and the Condition number of non-linear travel-time equations with respect to the known source parameter vector).
- Introduction: Please clarify better the novelty of your paper. Do you have introduced the PQ and PV factors in the IPSO algorithm as stated in Line 105-106?
Response 4:
Thank you very much for your question, the innovation is shown in the description of the answer to question 1. There is an error in the writing format of the text. Pe, PQ and PV represent the microseismic probability factor Fe, the microseismic importance factor FQ, and the effective range factor PV respectively, which have been introduced in Section 3 of the text. The three parameters proposed in this paper are improvements to the objective function, as shown in Eq. (9). Lines 95-96 in the text have been modified as shown below.
Based on the above problems, the microseismic probability factor Pe, the microseismic importance factor PQ, and the effective range factor PV are introduced according to the actual conditions of coal mines.
is revised to:
Based on the above problems, the microseismic probability factor Fe, the microseismic importance factor FQ, and the effective range factor FV are introduced according to the actual conditions of coal mines.
- Line 118: Again, as Comment 3. It is not clear what is the D-value theory. From your references, it seems more appropriate to refers to this method as D-optimal design theory.
Response 5:
Thank you very much for your question. As explained in Lines 121-128 in the text, the D-value represents the Determinant value of the source covariance matrix, so that the microseismic sensor arrangement with the smallest D value is the optimal solution. As in Eq. (2) in the text, the covariance Cx can be graphically explained by the confidence ellipsoid, that is, the eigenvalues ​​of the covariance matrix constitute the length of the principal axis of the confidence ellipsoid. Finding the station arrangement with the smallest volume of the ellipsoid is called the optimal design of D- value. The volume of the ellipsoid is proportional to the product of the covariance eigenvalues, that is, to the determinant of Cx. As shown in Eq. (3) in the text, det[Cx] is minimized, so as to satisfy the D-value optimization criterion. In this paper, based on the D-value optimization algorithm, the microseismic probability factor Fe, the microseismic importance factor FQ, and the effective range factor FV are added to enrich the objective function of sensor optimization. Since there are many optimization methods to solve this objective function, such as genetic algorithm, this paper proposes to use the improved particle swarm optimization algorithm to solve the optimization problem. Therefore, the theme of this paper is to study the optimal arrangement of sensors based on the improved particle swarm optimization algorithm.
- (1). Why do you add three times (x0-xi)2? What does it mean? Is it a mistake?
Response 6:
Thank you very much for your question, There is no error in Eq. (1). It is the arrival time equation between the microseismic source and the station. The microseismic source is H(t0, x0, y0, z0), and the ith station is Si(ti, xi, yi, zi).
- Line 139-141: It is not clear what the authors mean with “hypocenter points calculated…”. How do you calculate them?
Response 7:
Thanks a lot for your question. In Lines 129-131 in the text, "hypocenter points calculated" refers to the hypocenter of the hypocenter to be calculated in the monitoring area. During the calculation, the known location of the hypocenter point in the monitoring area is assumed. Use Eq. (3) to calculate the objective function values for all hypocenter points.
- Line 152: You write about geophones, even if you used seismometers in your application (Line 393). It could be useful to explain what kind of measurement instruments can be used.
Response 8:
Thank you very much for your question. In Line 93, the field of this text is the application of coal mine microseismic monitoring. We use a geophone, whoes response frequency range is 15 Hz to 1000 Hz, and the sensitivity is 43.3 V/m/sec. The geophone used in this paper is suitable for engineering scales of <1km and response frequency of 500 Hz, so it is suitable for mine microseismic monitoring.
- Line 175: What do you mean with monitoring range?
Response 9:
Thank you very much for your question, as described on Lines 163-170 in the text, The macroscopic requirement of network optimization is that the spatial geometry formed by the stations has good properties, that is, the effective monitoring range and the designed monitoring range are highly consistent [25]. To construct the monitoring range index of the station network from a quantitative point of view, it is necessary to define the effective monitoring range. The D-value method is used to calculate the theoretical positioning error under the network, and it is considered that the three-dimensional space with an error less than e (the value of e is set by the mine according to the needs of safety production, such as 50 m) is an effective monitoring area [17].
- Line 196: How do you evaluate the effective range factor?
Response 10:
Thank you very much for your question, as described on Lines 170-176 in the text, The design monitoring range is recorded as V0, the effective monitoring range of the station network is V1, and the overlapping area of the design monitoring range and the effective monitoring range is V'. The microseismic importance factor FV is introduced, and the mathematical definition is shown in formula (5).
(5)
- Line 213 and 215: the definition of gbest is repeated.
Response 11:
Thank you very much for your question, "and the global extreme value of the whole particle swarm is gbest." has been removed as indicated on Lines 202 - 203 in the text, as shown below.
The position of the ith particle in the population in n-dimensional space is expressed as xi=(xi1, xi2, ···, xin), and its velocity is vi=(vi1, vi2, ···, vin), and the global extreme value of the whole particle swarm is gbest.
is revised to:
The position of the ith particle in the population in n-dimensional space is expressed as xi=(xi1, xi2, ···, xin), and its velocity is vi=(vi1, vi2, ···, vin).
- (6): Consider dividing this equation into (6a) and (6b).
Response 12:
Thank you very much for your question, Lines 206 - 207 Eq. (6) in the text has been changed as shown below.
|
(6) |
is revised to:
(6a)
(6b)
- Lines 223-224: Equations (12) and (13) are missing.
Response 13:
Thank you very much for your question. Eq. (12) and (13) in Line 217 are incorrectly written, and have been revised to Eq. (7) and (8), as shown below.
The bacterial foraging algorithm [29] was introduced to the particle swarm optimization (PSO) algorithm, which will be added to the focused behavior and the tropism behavior, namely such as equations (12) and (13) to increase the randomness of particle movement and modify particle fitness value [30].
is revised to:
The bacterial foraging algorithm was introduced to the particle swarm optimization (PSO) algorithm, which will be added to the chemotaxis behavior and the elimination-dispersal behavior, namely such as equations (7) and (8) to increase the randomness of particle movement and modify particle fitness value [30].
- Line 259-260. Fix all the subscripts.
Response 14:
Thank you very much for your question, the subscripts in Lines 256-257 have been modified as shown below.
In the formula, (Cx)ij is the matrix Cx of the element (i, j). sxy is the epicenter error. sz is the focal depth error, and sxyz is the focal spatial error.
is revised to:
sxy is the epicenter error. sz is the focal depth error, and sxyz is the focal spatial error.
- Line 277: “the improved particle algorithm proposed in this paper”. Please, clarify if the algorithm already existed or if you have written it.
Response 15:
Thank you very much for your question. The improved particle swarm algorithm proposed in this paper is an improved algorithm proposed by the author in 2021. The algorithm has been written independently and has been successfully applied to the microseismic localization algorithm, as shown in [30]. This time, it was successfully applied to the optimal arrangement of the sensor.
- Lines 311-312. From Figure 2 it seems that the interval in Y direction is [0 200], not [0 500] how you wrote. Moreover, again in Y direction, the grid spacing is 50 m (You wrote “The grid spacing is 100”).
Response 16:
Thank you very much for your question, the language in the text is wrong, Lines 302-304 have been modified as shown below.
For the mining working face, we assume that the X direction interval of the monitoring model is [0 m, 500 m], the Y direction interval is [0 m, 200 m], and the elevation is 200 m. The key monitoring area range is [200 m, 500 m] in the X direction, [0 m, 500 m] in the Y direction, and the elevation is 200m. The grid spacing is 100 m, and the sensor arrangement elevation is 0m.
is revised to:
For the mining working face, we assume that the X direction interval of the monitoring model is [0 m, 100 m], the Y direction interval is [0 m, 50 m], and the elevation is 200 m. The key monitoring area range is [200 m, 500 m] in the X direction, [0 m, 200 m] in the Y direction, and the elevation is 200 m. The grid spacing is 100 m ´ 50 m, and the sensor arrangement elevation is 0 m.
- Lines 327-328: It is the first time I read about temperature in this paper. It is never mentioned before. The same symbol (capital letter T) is used in Eq.(1) referring to time. Maybe, you can adopt a lowercase “t” for time.
Response 17:
Thank you very much for your questions and suggestions, as you suggested, T in Lines 324-325 has been modified to t as shown below.
The initial temperature T0 = 97. The end temperature Tf = 89.9, and the temperature drop ratio P = 0.99.
is revised to:
The initial temperature t0 = 97. The end temperature tf = 89.9, and the temperature drop ratio l = 0.99.
- Line 156: It is preferable to refer to Equation as “Eq. (10)” not “formula”.
Response 18:
Thank you very much for your questions and suggestions. All “formulas” in the text are modified to “Equation”, as shown below.
- Lines 112-113:the shortest time Ti from the source H to the ith station can be described by equation (1).
is revised to:
the shortest time Ti from the source H to the ith station Si can be described by Eq. (1).
- Line 114-116:In the formula, x(t0, x0, y0, z0) is the time and three-dimensional coordinates of the microseismic source, respectively.
is revised to:
In the equation, x(t0, x0, y0, z0) is the time and three-dimensional coordinates of the microseismic source, respectively.
- Line 120-121: In formula (2), A is the calculated partial differential matrix with the corresponding earthquake arrival time, and k is a constant.
is revised to:
In Eq. (2), A is the calculated partial differential matrix with the corresponding earthquake arrival time, and k is a constant.
- Line 126-127: As shown in formula (3), det[Cx] is minimized, to satisfy the D-value optimization criterion.
is revised to:
As shown in Eq. (3), det[Cx] is minimized, to satisfy the D-value optimization criterion.
- Line 128-129: In the formula, ne is the number of hypocenter points calculated in the monitoring area.
is revised to:
In the equation, ne is the number of hypocenter points calculated in the monitoring area.
- Line 155-156: The microseismic probability factor in the whole mining area always satisfies the formula (4).
is revised to:
The microseismic probability factor in the whole mining area always satisfies Eq. (4).
- Line 172-174: The microseismic importance factor PV is introduced, and the mathematical definition is shown in formula (5).
is revised to:
The microseismic importance factor PV is introduced, and the mathematical definition is shown in Eq. (5).
- Line 202-205: The position of the ith particle in the population in n-dimensional space is expressed as xi=(xi1, xi2, ···, xin), and its velocity is vi=(vi1, vi2, ···, vin), and the global extreme value of the whole particle swarm is gbest. When finding these two extremes, update the speed and position with the following formula.
is revised to:
The position of the ith particle in the population in n-dimensional space is expressed as xi=(xi1, xi2, ···, xin), and its velocity is vi=(vi1, vi2, ···, vin). When finding these two extremes, update the speed and position with the following equation.
- Line 247: In the formula, PV is the effective range factor.
is revised to:
In the equation, FV is the effective range factor.
- Line 256: In the formula, (Cx)ij is the matrix Cx of the element (i, j).
is revised to:
In the equation, (Cx)ij is the matrix Cx of the element (i, j).
- Line 294-296: â‘ According to formula (10), the sensitivity contour map and the positioning error map of the current network are calculated, to preliminarily evaluate the feasibility of the network layout.
is revised to:
â‘ According to Eq. (10), the sensitivity contour map and the positioning error map of the current network are calculated, to preliminarily evaluate the feasibility of the network layout.
Line 349-350: The source error and sensitivity of different sensor arrangement schemes are calculated by formula (10).
is revised to:
The source error and sensitivity of different sensor arrangement schemes are calculated by Eq. (10).

Reviewer 2 Report
The paper tries to solve a practical problem: how to design optimally seismic monitoring network in coal mines. It uses quite sophisticated algorithm for optimization. However, in my opinion, the algorithm does not take into account one factor, which is important in seismic network design - attenuation of seismic waves. The determination of onset time at the seismic station is more precise (and more valuable for location) if the signal-to-noise ratio is high, that means, if the amplitude of seismic wave is big. That means, it is optimal to place the seismic stations as close to the source as possible. Of course, the geometry of the network, which is investigated in the paper, is also important, but the distance from the hypocenter is crucial. I think, this should be addressed in the paper.
I have several minor suggestions to the text:
53 location accuracy of microseisms - the term „microseisms“ has different meaning in seismology – noise with period about 6s. Please, use another term.
122 and next paragraphs – the letter X is used both for vector and one coordinate. It is confusing. Please, use another letter.
187 P2 is used instead of PV. Why?
192 “dynamic disasters” is an unusual term. Does it mean rockbursts?
211 “P” is used for Particle, but earlier it was used for effective range factor and for importance factor. Please, use different symbol.
374 “slope” should be stope
Author Response
Response Letter- Reviewer 2
First of all, the authors highly appreciate the editors and reviewers’ comments on the manuscript. We carefully reviewed the comments and suggestions and have incorporated major revisions into the manuscript. Hopefully, the revised version could satisfy the reviewers. We also welcome further comments and suggestions if they may have. Listed below are the authors’ point-by-point responses to the reviewers’ comments. The revisions are highlighted red in the revised manuscript.
- The paper tries to solve a practical problem: how to design optimally seismic monitoring network in coal mines. It uses quite sophisticated algorithm for optimization. However, in my opinion, the algorithm does not take into account one factor, which is important in seismic network design - attenuation of seismic waves. The determination of onset time at the seismic station is more precise (and more valuable for location) if the signal-to-noise ratio is high, that means, if the amplitude of seismic wave is big. That means, it is optimal to place the seismic stations as close to the source as possible. Of course, the geometry of the network, which is investigated in the paper, is also important, but the distance from the hypocenter is crucial. I think, this should be addressed in the paper.
Response 1:
Thank you very much for your comments. In this paper, when calculating the sensor optimization objective function module, the seismic wave attenuation is idealized and simplified, which is beneficial to improve the calculation efficiency. Thanks for your question suggestion, in future research work, we will increase the optimization effect of seismic wave attenuation. However, in the sensor arrangement evaluation module, we consider the attenuation effect of seismic waves, and calculate the sensitivity of the sensor arrangement scheme according to the propagation distance, as described in Lines 260-266. The expected standard deviation graph drawn by Eq. (10) is a function of event magnitude, that is, the equation represents the source location standard error whose magnitude is ML and whose source coordinates are Hi. In the proposed monitoring area, the event magnitude ML can be related to its measurable distance r. The distance r from the point Hi to the monitoring station can be calculated, and then the distance can be converted into an earthquake magnitude, and a sensitivity contour map can be drawn.
- 53 location accuracy of microseisms - the term „microseisms“ has different meaning in seismology – noise with period about 6s. Please, use another term.
Response 2:
Criticisms are greatly appreciated, Lines 41-42 have been revised in the text as shown below.
The quality of the network layout directly affects the location accuracy of microseisms [7-8].
is revised to:
The quality of the network layout directly affects microseismic location accuracy [7-8].
- 122 and next paragraphs – the letter X is used both for vector and one coordinate. It is confusing. Please, use another letter.
Response 3:
Thank you very much for your comments, X has been replaced with H and Si in Lines 111-116 as shown below.
The microseismic source H is x(t0, x0, y0, z0), and the ith station is xi(ti, xi, yi, zi). For the uniform and isotropic velocity models, the shortest time Ti from the source H to the ith station can be described by equation (1).
In the formula, x(t0, x0, y0, z0) is the time and three-dimensional coordinates of the microseismic source, respectively. xi(ti, xi, yi, zi) are the time and three-dimensional coordinates of the ith sensor, respectively.
is revised to:
The microseismic source is H(t0, x0, y0, z0), and the ith station is Si(ti, xi, yi, zi). For the uniform and isotropic velocity models, the shortest time Ti from the source H to the ith station Si can be described by Eq. (1).
In the equation, H(t0, x0, y0, z0) is the time and three-dimensional coordinates of the microseismic source, respectively. Si(ti, xi, yi, zi) are the time and three-dimensional coordinates of the ith sensor, respectively.
- 187 P2 is used instead of PV. Why?
Response 4:
Thank you very much for your questions and comments. The writing of P2 in the text is wrong. The P2 in Line 175-176 has been changed to FV. The text is as follows.
Ideally, the effective monitoring range of the station network completely coincides with the designed monitoring range, that is, P2=1, and it is stipulated that P2<=1.
is revised to:
Ideally, the effective monitoring range of the station network completely coincides with the designed monitoring range, that is, FV=1, and it is stipulated that FV<=1.
- 192 “dynamic disasters” is an unusual term. Does it mean rockbursts?
Response 5:
Thank you very much for your comments, Lines 180-181 "dynamic disasters" refers to “coal or rock dynamic disasters”, the text has been changed to "coal or rock dynamic disasters" as shown below.
and prevent the occurrence of dynamic disasters.
is revised to:
and prevent the occurrence of coal or rock dynamic disasters.
- 211 “P” is used for Particle, but earlier it was used for effective range factor and for importance factor. Please, use different symbol.
Response 6:
Thank you very much for your comments. All the parameters of PQ, PV and Pe in the text have been modified to FQ, FV, Fe, as follows.
- Lines 11-12: the microseismic probability factor Pe, the microseismic importance factor PQ, and the effective range factor PV are introduced to establish the objective function of the optimal network layout.
is revised to:
Introducing the microseismic probability factor Fe, the microseismic importance factor FQ, and the effective range factor FV,
- Lines 94-96: Based on the above problems, the microseismic probability factor Pe, the microseismic importance factor PQ, and the effective range factor PV are introduced according to the actual conditions of coal mines.
is revised to:
Based on the above problems, the microseismic probability factor Fe, the microseismic importance factor FQ, and the effective range factor FV are introduced according to the actual conditions of coal mines.
- Lines 242-244: the microseismic probability factor Pe, the microseismic importance factor PQ, and the effective range factor PV are introduced to establish the objective function of the optimal network layout scheme.
is revised to:
the microseismic probability factor Fe, the microseismic importance factor FQ, and the effective range factor FV are introduced to establish the objective function of the optimal network layout scheme.
- Lines 247-248: In the formula, PV is the effective range factor. PQj is the importance factor in the jth area of the microseismic monitoring space. Pe is the microseismic probability factor.
is revised to:
In the equation, FV is the effective range factor. FQj is the importance factor in the jth area of the microseismic monitoring space. Fe is the microseismic probability factor.
- Lines 286-288: determine the microseismic probability factor Pe, the microseismic importance factor PQ, and the effective range factor PV, to form a set of station candidate points.
is revised to:
determine the microseismic probability factor Fe, the microseismic importance factor FQ, and the effective range factor FV, to form a set of station candidate points.
- Lines 312-313: PQ of the key monitoring area is 1.5, and PQ of other monitoring areas is 1.0.
is revised to:
FQ of the key monitoring area is 1.5, and FQ of other monitoring areas is 1.0.
- Lines 359-361: In addition, it can be obtained by calculation that the effective range factor PV of scheme I is 0.44, the effective range factor PV of scheme II is 0.75, and the effective range factor PV of scheme III is 0.87.
is revised to:
In addition, it can be obtained by calculation that the effective range factor FV of scheme I is 0.44, the effective range factor FV of scheme II is 0.75, and the effective range factor FV of scheme III is 0.87.
- 374 “slope” should be stope.
Response 7:
Thank you very much for your comments, the slope in Linew 369-371 has been changed to stope, as shown below.
The minefield is 4 kilometers long, with a slope width of 3.3 kilometers, and a coal-bearing area of 13.2 square kilometers.
is revised to:
The minefield is 4 kilometers long, with a stope width of 3.3 kilometers, and a coal-bearing area of 13.2 square kilometers.

Reviewer 3 Report
The authors have come up a novel optimization algorithm for optimal placement of sensors in a microseismic monitoring of a coal mine with specific application to Xiashijie in Tongchuan, Shaanxi. Both the optimal scheme and its evaluation for station arrangement is presented with agreeable levels of accuracy, even compared to the state-of-the-art algorithms in the literature. This work can be accepted as it is.
Author Response

(The authors gave the same response as above.)

Round 2
Reviewer 2 Report
The authors have improved the paper a lot after revision, my congratulation. I found only one misprint:
395 microseismic importance factor PQ
It should be FQ
Author Response
Thank you very much for your acknowledgment and your objective evaluation of this article. The corresponding content in the text has been modified according to your suggestion. In Line 395, PQ has been revised to FQ. The modifications in the text are as follows.
and the microseismic importance factor PQ of the key monitoring area is 1.5.
is revised to
and the microseismic importance factor FQ of the key monitoring area is 1.5.
